# Socioeconomic and contextual correlates of suicidal ideation among Indonesian adults: Evidence from a multilevel analysis of the 2018 National Health Survey

Sujarwoto Sujarwoto[1], Penny Bee[2], Helen Brooks[2], Asri Maharani [2]*

**1** Department of Public Administration, Universitas Brawijaya, Malang, East Java, Indonesia, **2** Division of Nursing, Midwifery and Social Work, School of Health Sciences, University of Manchester, Manchester, Lancashire, United Kingdom

* asri.maharani@manchester.ac.uk

## Abstract

Suicide is a major public health concern and a leading cause of premature death, particularly in low- and middle-income countries. In Indonesia, the true burden is likely underestimated due to stigma and underreporting. Evidence on suicidal ideation, an important precursor to suicide, remains limited. This study aims to identify individual and community correlates of persistent suicidal ideation among Indonesian adults using nationally representative data. A cross-sectional analysis was conducted using data from the 2018 Indonesia National Health Survey covering adults aged 18 years and older. District-level indicators were obtained from the 2018 Village Potential Statistics and regional economic data from the National Bureau of Statistics. Multivariable and multilevel logistic regression models identified individual and contextual factors associated with suicidal ideation, accounting for individuals nested within districts. Among 636,285 adults (mean age 42.5 years; 47.4% female), 0.87% reported persistent suicidal ideation in the past month. Higher community social capital (OR = 0.94; 95% CI 0.88–1.00) and district GDP (OR = 0.91; 95% CI 0.86–0.97) were associated with lower odds, while social deprivation increased risk (OR = 1.13; 95% CI 1.06–1.21). Lower odds of suicidal ideation were observed among women (OR = 0.56), married (OR = 0.65) or widowed individuals (OR = 0.77), those with higher education attainments (OR = 0.32 for university graduates), and resident of Java (OR = 0.61). Divorce (OR = 1.28), older age (OR = 1.83 for ages 65–74), chronic illness (OR = 1.56 for heart disease), and poor self-rated health (OR = 3.38) were linked to higher risk. Strengthening community social capital and reducing social deprivation are vital to prevent suicidal ideation in Indonesia. Interventions should address socioeconomic inequalities and improve access to health and social support, especially among older adults and those with chronic illness. Promoting inclusive economic growth and community resilience may help mitigate the underlying stressors contributing to suicidal thoughts.

**Data availability statement:** Data are available upon official request. The data set (RISKESDAS) can be accessed with approval by Director General of Health Policy Agency, Ministry of Health, Republic of Indonesia, at https://layanandata.kemkes.go.id/request. The authors had no special access privileges; others may obtain the data in the same manner. Community indicators from the 2018 Village Potential Statistics (PODES) from Statistics Indonesia (BPS) are available from Statistics Indonesia following registration (https://ppid.bps.go.id/?mfd=0000). All variables used in this study are described in the article and Supporting information.

**Funding:** The author(s) received no specific funding for this work.

**Competing interests:** The authors have declared that no competing interests exist.

## Introduction

Suicidality, encompassing suicidal thoughts, plans, and behaviours, represents a major global public health concern. It causes hundreds of thousands of deaths each year, surpassing mortality from malaria, HIV/AIDS, breast cancer, and armed conflict, and imposes profound social, emotional, and economic burdens on families and communities [1,2]. In recognition of its wide-ranging impact, the World Health Organization (WHO) has made suicide prevention a global priority within the Sustainable Development Goals (SDGs), its General Programme of Work, and the Live Life framework, which promotes evidence-based interventions across individual, community, and policy levels [2]. A comprehensive, coordinated prevention strategy is therefore essential to protect lives and mitigate the long-term consequences of suicidality for both individuals and society.

According to WHO estimates, Indonesia experiences around 6,500 suicides each year [2]. This figure is likely an underestimate, as strong stigma surrounding mental illness and suicide discourages families from reporting deaths and leads to their classification under other causes [1]. Recent analyses suggest that suicide underreporting may reach as high as 85.9%, even when enhanced surveillance and verbal autopsy methods are used, indicating that official statistics capture only about half of all cases [3]. The highest suicide rates were recorded in Bali, the Riau Islands, Yogyakarta, Central Java, and Central Kalimantan. Men had approximately twice the suicide rate of women. Hanging and self-poisoning were the most common methods, with rural suicide rates exceeding urban rates by more than fourfold, reflecting disparities in mental health service access and social isolation in remote areas [3,4].

Research on suicidality, including its spectrum from suicidal thoughts to attempts and deaths, has largely focused on high-income countries, leaving major evidence gaps in low- and middle-income countries (LMICs) where social, economic, and health contexts differ substantially [2,5]. This study specifically examines suicidal ideation as a critical yet underexplored component of suicidality. While suicidal ideation often reflects early stages of psychological distress and constitutes a key risk factor for subsequent suicidal behaviour, the relationship between ideation and action is complex and not necessarily linear. Many individuals who experience suicidal thoughts do not progress to suicidal behaviour, and multiple psychological, social, and contextual factors may either exacerbate or buffer this trajectory. Understanding its determinants is essential for prevention efforts in resource-limited settings. Evidence from high-income contexts suggests that community social capital can protect against suicidal ideation by enhancing belonging, social support, and mutual aid [6,7]. Community social capital, defined as the strength of social networks, shared norms, and trust that facilitate collective action, emerged as a protective contextual factor associated with lower odds of suicidal ideation. Conversely, low social capital and social fragmentation may heighten vulnerability through isolation, diminished collective efficacy, and reduced access to help [8,9].

However, these associations are rarely tested in LMIC contexts, where the social meanings of connectedness and community support may differ. In Indonesia, where suicide remains highly stigmatised and frequently underreported, the complex

interplay between mental health conditions, socioeconomic hardship, and community resources such as social capital has received limited empirical attention. Drawing on Durkheim's sociological theory of suicide [10], which highlights the protective role of social integration and connectedness, social capital offers a framework for understanding how collective ties and social cohesion influence vulnerability to suicidality. Empirical research in LMICs supports this theoretical perspective: studies in China and India have shown that stronger family cohesion, community participation, and perceived social support are associated with lower suicidal ideation and attempts [11–13]. A large Chinese university survey found that higher social capital was associated with lower suicidal ideation among medical students, and also helped buffer the impact of uncertainty stress [14]. It is thus crucial for understanding how community-level resilience and deprivation shape vulnerability to suicidality in rapidly changing LMIC contexts.

The present study aims to investigate how personal and contextual factors, including education, income, health status, community social capital, social deprivation, and district-level economic conditions, correlate with suicidal ideation among adults aged 18 years and older. The overarching research question guiding this study is: "What individual and community determinants are associated with suicidal ideation among Indonesian adults?"

This study makes several novel contributions. First, it provides the first multilevel national evidence on suicidal ideation in Indonesia, integrating data from the 2018 National Health Survey (Riskesdas) and the Village Potential Statistics (PODES). Second, it expands the understanding of suicidality in LMICs by linking social and economic contexts, particularly community social capital and social deprivation, to individual mental health outcomes [7,9]. Finally, the study contributes policy-relevant insights for designing community-based suicide prevention strategies that strengthen social capital and reduce contextual deprivation in resource-constrained settings.

## Materials and methods

### Study design

This cross-sectional analysis utilised data from the 2018 Indonesia National Health Survey (Riskesdas), a nationally representative survey conducted quinquennially [15]. Riskesdas, managed by the National Institute of Health Research and Development (NIHRD), includes health indicators such as mental health and non-communicable diseases. Ethical approval was obtained from the NIHRD prior to data collection, and informed written consent was obtained from all participants.

A multistage systematic random sample technique was employed to identify 636,285 persons aged 18 and above with comprehensive suicidal ideation data [15]. Hence, the study included individuals with complete information on key variables related to suicidal ideation, socioeconomic characteristics, and community-level indicators. Cases with missing data on these variables were excluded using listwise deletion, resulting in consistent analytic samples across models. This approach ensured internal validity while maintaining comparability with the national sampling frame. Descriptive characteristics and model estimates, therefore, reflect weighted data from respondents meeting these inclusion criteria.

The Riskesdas data were correlated with the 2018 Village Survey (PODES) to obtain village-level insights into community social capital and social deprivation, alongside district gross domestic product (GDP) data for 2017 from the Indonesian Bureau of Statistics [16]. Data were consolidated at the district level and linked to individual Riskesdas data, accounting for the hierarchical nature of individuals within districts.

### Ethics statement

This study used secondary, de-identified data from the 2018 Indonesia Basic Health Survey (Riset Kesehatan Dasar, Riskesdas), which was conducted by the National Institute of Health Research and Development (NIHRD), Ministry of Health, Indonesia. The Riskesdas survey protocols were reviewed and approved by the Health Research Ethics Commission of the NIHRD (Ethical Approval No. LB.02.01/2/KE.267/2017). Written informed consent was obtained from all participants by the NIHRD at the time of data collection. The present analysis was approved by the data custodians and

conducted in accordance with relevant national and institutional ethical guidelines. As this study involved secondary analysis of anonymised data, no additional ethical approval was required from the authors' institutions.

## Measure of suicidal ideation

Suicidal ideation was assessed using a single self-report item from the Riskesdas 2018. Respondents were asked: *"During the past two weeks, have you had recurrent thoughts of hurting yourself, wanting to kill yourself, or wishing that you were dead?"* Responses were coded as 1 for "yes" and 0 for "no". The two-week recall period and emphasis on recurrent thoughts indicate that this item captures persistent suicidal ideation or death wishes, rather than transient or occasional suicidal thoughts. Single-item measures of this type are commonly used in large population-based surveys to identify individuals experiencing sustained suicidal distress, although they do not capture the full multidimensionality of suicidal ideation assessed by comprehensive clinical instruments.

## Individual-level covariates

Individual-level variables included demographic, socioeconomic, health, and behavioural characteristics. Demographic factors comprised age, sex, and marital status. Age was grouped into seven categories (18–24, 25–34, 35–44, 45–54, 55–64, 65–74, and 75 years or older), with the 18–24 age group as the reference category. Sex was coded with males as the reference group. Marital status was classified as never married, married, divorced, or widowed.

Socioeconomic characteristics included educational attainment, household expenditure, and employment status. Education was divided into seven categories, ranging from no schooling to college or university level, with higher education serving as the reference. Household expenditure was grouped into quintiles, representing relative economic standing. Employment status captured diverse labour categories, including unemployed, student, civil servant, army or police officer, private-sector employee, self-employed, farmer, fisherman, driver or household assistant, and other occupations, with unemployed individuals serving as the reference group.

Health-related variables included both self-reported chronic conditions and perceived health. Respondents were asked whether they had ever been diagnosed by a health professional with conditions, including joint disease, hypertension, stroke, diabetes, heart disease, asthma, cancer, and renal failure. Self-rated health was assessed on a three-point scale: good, adequate, or poor. Health behaviours encompassed smoking and alcohol consumption. Smoking status was classified as daily smoker, occasional smoker, former smoker, or non-smoker. Alcohol consumption was grouped as none, under standard, or above standard. Finally, household and geographical characteristics were considered. Household expenditure quintiles captured relative economic position, and the island of residence distinguished between respondents living in Java and those residing in other islands of Indonesia.

## District-level covariates

Community social capital was measured through principal component analysis of four indicators: the frequency of communal labour activities ("*gotong royong*"), the number of community-based financial institutions, the frequency of social support initiatives, and the number of community representative organisations over the past three years. This approach follows prior ecological studies linking community engagement and institutional density to mental health and suicide prevention outcomes [7,17]. Social deprivation was assessed using principal component analysis of crime rates and community conflicts during the same period, reflecting previous conceptualisations of area deprivation and psychosocial stressors [8,9]. District-level GDP data from 2017 were included as a proxy for local economic development, consistent with earlier evidence on the protective role of economic opportunity against suicidality [18,19]. Although GDP data precede the *Riskesdas* survey by one year, this temporal lag is unlikely to bias results because district-level socioeconomic indicators in Indonesia are highly stable from year to year, reflecting gradual economic and demographic change rather than abrupt

shifts. The use of the most recent pre-survey GDP data, therefore, provides a valid approximation of the district economic context at the time of data collection.

## Statistical analysis

We first mapped the geographic distribution of suicidal ideation at the district level to visualise spatial variation. Descriptive statistics were then used to summarise individual and contextual characteristics, and bivariate associations were examined using chi-squared tests for categorical variables.

To identify factors associated with suicidal ideation, we employed multivariable and multilevel logistic regression models. The multilevel approach accounted for the hierarchical data structure, with individuals (Level 1) nested within districts (Level 2). Model 1 included individual-level variables (demographic, socioeconomic, behavioural, and health factors), while Model 2 additionally incorporated district-level indicators of community social capital, social deprivation, and GDP. This structure allowed estimation of both within-district and between-district effects on suicidal ideation.

All analyses accounted for the complex survey design of the 2018 Indonesia National Health Survey, which involved stratification, clustering, and sampling weights. Prior to modelling, individual-level sampling weights provided within the Indonesia National Health Survey data were applied to ensure representativeness at the provincial and district levels. Survey weights were scaled to the primary sampling unit (PSU) level following Stata's svyset procedures, enabling adjustment for unequal probabilities of selection across strata. In multilevel models, the scaled weights were incorporated at the individual level, while district-level predictors captured contextual variance. This approach ensures that estimated standard errors and confidence intervals reflect Indonesia's multistage sampling design and hierarchical population structure.

Model diagnostics included examination of multicollinearity using variance inflation factors (VIF < 2) and assessment of model fit using the intraclass correlation coefficient (ICC) and Akaike Information Criterion (AIC). Marginal effects were computed as predicted probabilities of suicidal ideation derived from the final multilevel logistic regression model. These probabilities were estimated using Stata's *margins* command, holding other covariates at their mean values. This approach enables intuitive interpretation of the strength and direction of associations between district-level predictors, including social capital, social deprivation, and GDP, and the likelihood of suicidal ideation, while accounting for the nested data structure. All analyses were conducted in Stata version 18.0, with a two-tailed significance threshold of $p < 0.05$.

## Results

### Characteristics of participants

The characteristics of all participants are presented in S1 Table. Of the 636,285 respondents, 52.6% were male and 47.4% female. Most participants (75.1%) were married, with a mean age of 42.3 years, and 27.9% had completed senior high school. Approximately 28.6% were unemployed, while 25.5% worked in agriculture. Joint problems (8.5%) and hypertension (8.8%) were the most frequently reported chronic conditions. Daily smoking was reported by 27.1%, and 2.0% described their health as suboptimal. Over half of the sample resided within the lower three quartiles of household expenditure.

The mean district-level social capital score was 0.14 (range from 1.34 to 5.68), and the social deprivation index averaged 0.13 (range from 1.06 to 6.65). These standardised indices were derived using principal component analysis, with higher values indicating stronger community engagement and collective participation for social capital, and greater exposure to crime and conflict for social deprivation. The logarithmic GDP score averaged 9.43 (range from 4.99 to 10.03), representing variation in district-level economic output per capita. Together, these variables capture contextual dimensions of community cohesion, social stress, and economic capacity relevant to mental health outcomes.

A total of 5,507 individuals (0.87%; 95% CI 0.84–0.89%) reported persistent suicidal ideation in the past month. Persistent suicidal thoughts were most common among married males aged 35–54 with primary education who were

unemployed or employed as farmers, often reporting hypertension or asthma, belonging to economically disadvantaged households, and living outside Java (Table 1).

### Geographical distribution of suicide ideation

Geographical disparities in persistent suicidal ideation are apparent among Indonesian districts (Fig 1). The coastal areas of northern Papua, central and northern Sulawesi, northern and central Kalimantan, eastern Nusa Tenggara, northeastern and southeastern Sumatra, Bali, and southern West Java demonstrate the highest incidence (>4%).

### Multivariable logistic regression

Table 2 displays the outcomes of multivariable and multilevel logistic regression analyses. The female gender exhibited lower odds of persistent suicidal thoughts in comparison to the male gender (OR = 0.560, 95% CI 0.511–0.614). In comparison to never-married persons, married and widowed individuals exhibited lower odds (OR = 0.646, 95% CI 0.584–0.715 for married; OR = 0.769, 95% CI 0.670–0.883 for widowed), but divorced individuals showed increased odds of suicidal thoughts (OR = 1.284, 95% CI 1.094–1.507). The likelihood of persistent suicidal ideation increased with age, with the highest odds observed among individuals aged 65–74 years (OR = 1.831, 95% CI 1.566–2.140). Higher education was significantly correlated with reduced suicidal ideation. Employed individuals (civil servants, private sector workers, self-employed individuals, and farmers) demonstrated a reduced likelihood of persistent suicidal ideation in comparison to jobless individuals. Chronic illness correlated with an increased likelihood of persistent suicidal thoughts. Intermittent smokers and former smokers exhibited greater probabilities than daily smokers, whereas non-smokers demonstrated reduced risks. Non-drinkers had a reduced likelihood of suicidal thoughts compared to individuals drinking below the standard limit. Individuals with poor self-rated health had markedly higher odds of suicidal ideation (OR = 10.38, 95% CI 9.50–11.33), consistent with a strong positive association between perceived ill-health and suicidal thoughts. Increased family expenditure correlated with diminished odds. Residing in Java was correlated with a reduced likelihood compared to residing in other locations.

### Multilevel logistic regression

The correlations at the individual level with persistent suicidal thoughts were predominantly stable. Community social capital and elevated GDP were significantly correlated with diminished odds of suicidal ideation (OR=0.935, 95% CI 0.875–1.00 for community social capital; OR=0.909, 95% CI 0.856–0.966 for district GDP), while higher levels of social deprivation were significantly associated with increased odds of suicidal ideation (OR = 1.13; 95% CI 1.057–1.209).

### Margins effect

Fig 2 presents the predicted probabilities of persistent suicidal ideation derived from the final multilevel logistic regression model. The results demonstrate that higher levels of community social capital and district GDP were associated with a lower predicted probability of suicidal ideation, whereas increasing social deprivation corresponded to higher predicted probabilities. This pattern is consistent with the multilevel regression results, showing that higher levels of community social capital and district economic development were associated with lower odds of persistent suicidal ideation, whereas greater social deprivation was associated with higher odds.

## Discussion

### Hidden burden of suicidal ideation in Indonesia

This national, population-based study examined the individual and community factors associated with suicidal ideation among Indonesian adults using linked survey and contextual data. The prevalence of persistent suicidal ideation was

**Table 1. Characteristics of participants with persistent suicidal ideation.**

| Variables | N obs. | % or mean | SD |
|---|---|---|---|
| Sex | | | |
| Male | 3,418 | 62.1% | 48.5% |
| Female | 2,089 | 37.9% | 48.5% |
| Marital status | | | |
| Never married | 802 | 14.6% | 35.3% |
| Married | 3,759 | 68.3% | 46.6% |
| Divorced | 255 | 4.6% | 21.0% |
| Widowed | 691 | 12.5% | 33.1% |
| Age (Mean) | | 45.52 | 16.26 |
| Age group | | | |
| 18-24 | 650 | 11.8% | 32.3% |
| 25-34 | 889 | 16.1% | 36.8% |
| 35-44 | 1,135 | 20.6% | 40.5% |
| 45-54 | 1,189 | 21.6% | 41.1% |
| 55-64 | 910 | 16.5% | 37.1% |
| 65-74 | 489 | 8.9% | 28.4% |
| >=75 | 245 | 4.4% | 20.6% |
| Education | | | |
| Never go to school | 668 | 12.1% | 32.7% |
| Less than primary school | 1,211 | 22.0% | 41.4% |
| Primary school | 1,604 | 29.1% | 45.4% |
| Junior high school | 845 | 15.3% | 36.0% |
| Senior high school | 982 | 17.8% | 38.3% |
| Diploma | 77 | 1.4% | 11.7% |
| College/University | 120 | 2.2% | 14.6% |
| Employment status | | | |
| Unemployed | 2,077 | 37.7% | 48.5% |
| Students | 137 | 2.5% | 15.6% |
| Civil servant/army/police officer | 83 | 1.5% | 12.2% |
| Private worker | 189 | 3.4% | 18.2% |
| Self-employed | 571 | 10.4% | 30.5% |
| Farmer | 1,585 | 28.8% | 45.3% |
| Fisherman | 84 | 1.5% | 12.3% |
| Driver/household assistance | 432 | 7.8% | 26.9% |
| Others | 349 | 6.3% | 24.4% |
| Joint diseases | | | |
| No | 4,605 | 83.6% | 37.0% |
| Yes | 902 | 16.4% | 37.0% |
| Hypertension | | | |
| No | 4,692 | 85.2% | 35.5% |
| Yes | 815 | 14.8% | 35.5% |
| Stroke | | | |
| No | 5,279 | 95.9% | 19.9% |
| Yes | 228 | 4.1% | 19.9% |
| Diabetes | | | |
| No | 5,255 | 95.4% | 20.9% |
| Yes | 252 | 4.6% | 20.9% |

*(Continued)*

**Table 1.** (Continued)

| Variables | N obs. | % or mean | SD |
|---|---|---|---|
| Heart diseases | | | |
| No | 5,252 | 95.4% | 21.0% |
| Yes | 255 | 4.6% | 21.0% |
| Asthma | | | |
| No | 5,160 | 93.7% | 24.3% |
| Yes | 347 | 6.3% | 24.3% |
| Cancer | | | |
| No | 5,468 | 99.3% | 8.4% |
| Yes | 39 | 0.7% | 8.4% |
| Renal failure | | | |
| No | 5,432 | 98.6% | 11.6% |
| Yes | 75 | 1.4% | 11.6% |
| Smoking status | | | |
| Smoking every day | 1,326 | 24.1% | 42.8% |
| Smoking not every day | 305 | 5.5% | 22.9% |
| Ex smoker | 355 | 6.4% | 24.6% |
| Not smoker | 3,521 | 63.9% | 48.0% |
| Alcohol consumption | | | |
| Under standard | 243 | 4.4% | 20.5% |
| More than standard | 170 | 3.1% | 17.3% |
| No alcohol | 5,094 | 92.5% | 26.3% |
| Self-rated health | | | |
| Good | 2,610 | 47.4% | 49.9% |
| Adequate | 2,065 | 37.5% | 48.4% |
| Poor | 832 | 15.1% | 35.8% |
| Household expenditure | | | |
| 1st quartile | 1,245 | 22.6% | 41.8% |
| 2nd | 1,234 | 22.4% | 41.7% |
| 3rd | 1,122 | 20.4% | 40.3% |
| 4th | 1,067 | 19.4% | 39.5% |
| 5th | 839 | 15.2% | 35.9% |
| Island | | | |
| Outer Java | 4,154 | 75.4% | 43.1% |
| Java | 1,353 | 24.6% | 43.1% |

0.87% (95% CI 0.84–0.89%), representing 5,507 individuals. By integrating data from both household and district levels, this study addressed its primary aim, to assess how socioeconomic, health, and contextual factors, including social capital, social deprivation, and local economic development, are associated with suicidal thoughts in Indonesia.

The observed prevalence is lower than that reported in many low- and middle-income as well as high-income countries, such as China (3.9%) [20], South Korea (15.2%) [21], the United States (3.2%) [22], and India (11.0%) [23]. However, this comparatively low rate should be interpreted with caution due to the high likelihood of underreporting related to stigma, religious prohibition, and social desirability bias [1,3]. These cultural factors often lead to concealment or misclassification of suicidality as other causes of death or illness, particularly in conservative communities. Thus, the actual burden of suicidal ideation in Indonesia is likely higher than captured in national surveys.

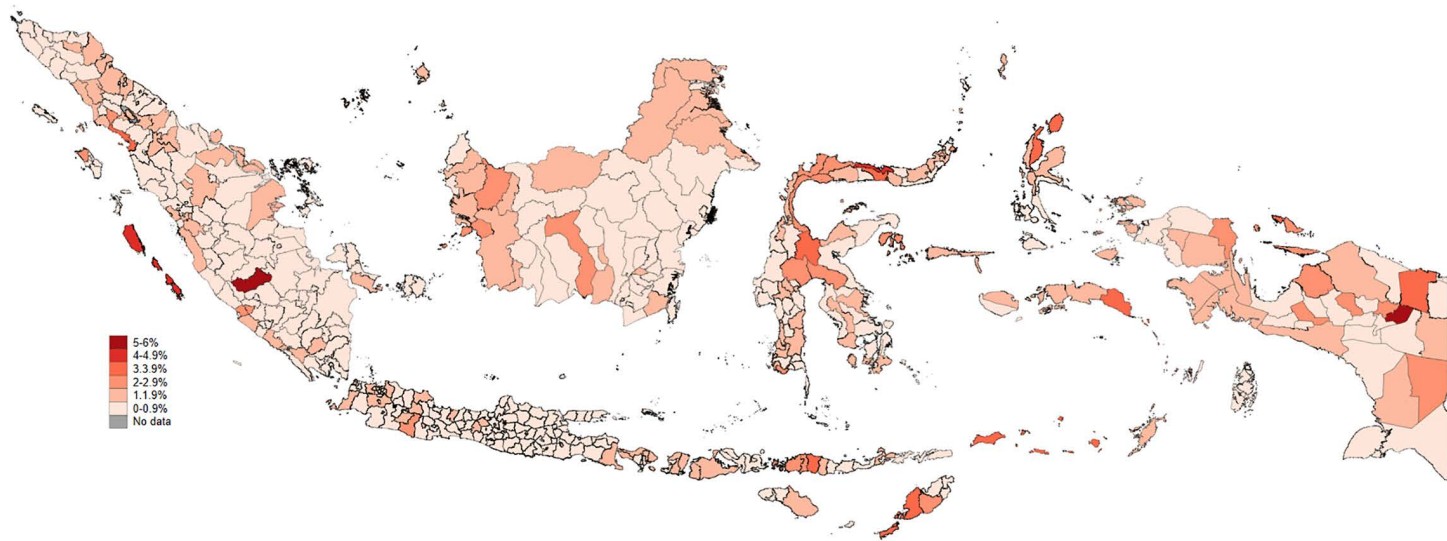

**Fig 1. Geographical distribution of suicidal ideation.** Map redrawn by the authors using openly licensed geographic boundary data from openfreemap.org (**https://openfreemap.org/**); the figure is distributed under the Creative Commons Attribution (CC BY 4.0) licence.

When contextualised globally, Indonesia's prevalence suggests a lower recorded burden but a substantial hidden risk. This aligns with patterns observed across LMICs where stigma and weak mental-health surveillance systems suppress disclosure. Studies from Europe, Australia, and Canada also show considerable variation, influenced by socioeconomic inequalities and access to care [24–26]. Therefore, while Indonesia's reported prevalence appears modest, it underscores the urgent need to strengthen mental-health literacy, destigmatise help-seeking, and improve surveillance accuracy.

## Community contexts: Social capital, deprivation, and economic opportunity

We found that community social capital was significantly correlated with lower odds of persistent suicidal ideation. Evidence from high-income and Asian contexts consistently links stronger community cohesion with reduced suicide mortality and ideation rates [6,7,17,27]. In Indonesia, communities with higher participation in collective activities and stronger interpersonal trust may foster a sense of belonging and mutual support, thereby buffering the effects of stress and isolation as major risk factors for suicidality. Empirical evidence from Indonesia suggests that neighbourhood-level trust and collective engagement can be protective for mental health, including lower risks of depressive symptoms, particularly when community participation is experienced as supportive and reciprocal [28]. These findings resonate with Durkheim's sociological theory of suicide [10], which emphasises the protective function of social integration and regulation in maintaining psychological stability. However, the mental health implications of community social capital are not uniformly positive. In the Indonesian context, collective practices such as *gotong royong* may also entail obligation and informal enforcement of village norms rather than voluntary engagement. Prior scholarship has noted that the institutionalisation of gotong royong can transform a practice of mutual aid into a compulsory form of participation, thereby introducing elements of social control and pressure [29]. Excessively rigid or coercive forms of social regulation may generate psychological strain rather than protection. As such, high levels of collective participation may simultaneously signal social integration while exposing individuals, especially those who are socially or economically marginalised, to stress or sanctions.

The relationship between district-level GDP and suicidal ideation was inverse: higher GDP was associated with lower odds of persistent suicidal ideation. Economic prosperity often coincides with better access to health and social services, reduced financial stress, and enhanced community infrastructure, all of which may support mental well-being [18,19].

**Table 2. Multivariable and multilevel logistic regression results.**

| Variables | Model 1 | | | | Model 2 | | | |
|---|---|---|---|---|---|---|---|---|
| | Odds ratio | P-Value | 95% Confidence intervals | | Odds ratio | P-Value | 95% Confidence intervals | |
| | | | Lower | Upper | | | Lower | Upper |
| **Individual-level variables** | | | | | | | | |
| Female | 0.560 | 0.000 | 0.511 | 0.614 | 0.583 | 0.000 | 0.532 | 0.639 |
| Marital status, reference: never married | | | | | | | | |
| Married | 0.646 | 0.000 | 0.584 | 0.715 | 0.623 | 0.000 | 0.563 | 0.691 |
| Divorced | 1.284 | 0.002 | 1.094 | 1.507 | 1.292 | 0.002 | 1.100 | 1.518 |
| Widowed | 0.769 | 0.000 | 0.670 | 0.883 | 0.754 | 0.000 | 0.656 | 0.866 |
| Age group (years), reference: 18–24 | | | | | | | | |
| 25-34 | 0.983 | 0.774 | 0.873 | 1.106 | 1.003 | 0.964 | 0.891 | 1.129 |
| 35-44 | 0.959 | 0.506 | 0.849 | 1.084 | 0.982 | 0.770 | 0.868 | 1.110 |
| 45-54 | 0.969 | 0.624 | 0.854 | 1.099 | 1.013 | 0.848 | 0.891 | 1.150 |
| 55-64 | 1.117 | 0.107 | 0.976 | 1.279 | 1.179 | 0.018 | 1.029 | 1.352 |
| 65-74 | 1.831 | 0.000 | 1.566 | 2.140 | 1.923 | 0.000 | 1.642 | 2.253 |
| >=75 | 1.813 | 0.000 | 1.504 | 2.184 | 1.933 | 0.000 | 1.600 | 2.335 |
| Education, reference: no school | | | | | | | | |
| Less than primary school | 0.928 | 0.132 | 0.842 | 1.023 | 0.938 | 0.209 | 0.849 | 1.037 |
| Primary school | 0.719 | 0.000 | 0.654 | 0.791 | 0.717 | 0.000 | 0.649 | 0.791 |
| Junior high school | 0.587 | 0.000 | 0.525 | 0.656 | 0.597 | 0.000 | 0.533 | 0.670 |
| Senior high school | 0.463 | 0.000 | 0.414 | 0.519 | 0.465 | 0.000 | 0.414 | 0.523 |
| Diploma | 0.379 | 0.000 | 0.295 | 0.488 | 0.385 | 0.000 | 0.299 | 0.496 |
| College/University | 0.320 | 0.000 | 0.256 | 0.398 | 0.316 | 0.000 | 0.253 | 0.395 |
| Employment status, reference, unemployed | | | | | | | | |
| Students | 0.883 | 0.199 | 0.731 | 1.067 | 0.864 | 0.132 | 0.714 | 1.045 |
| Civil servant/army/police officer | 0.718 | 0.008 | 0.561 | 0.918 | 0.705 | 0.006 | 0.551 | 0.902 |
| Private worker | 0.661 | 0.000 | 0.565 | 0.775 | 0.684 | 0.000 | 0.583 | 0.803 |
| Self-employed | 0.814 | 0.000 | 0.738 | 0.899 | 0.848 | 0.001 | 0.767 | 0.937 |
| Farmer | 0.839 | 0.000 | 0.779 | 0.904 | 0.824 | 0.000 | 0.762 | 0.891 |
| Fisherman | 0.971 | 0.799 | 0.773 | 1.219 | 0.876 | 0.260 | 0.695 | 1.103 |
| Driver/household assistance | 0.935 | 0.241 | 0.835 | 1.046 | 0.961 | 0.496 | 0.857 | 1.077 |
| Others | 0.893 | 0.057 | 0.795 | 1.003 | 0.874 | 0.027 | 0.777 | 0.985 |
| Diagnosed with joint disease | 1.443 | 0.000 | 1.337 | 1.558 | 1.448 | 0.000 | 1.339 | 1.565 |
| Diagnosed with hypertension | 1.049 | 0.270 | 0.964 | 1.141 | 1.052 | 0.241 | 0.967 | 1.145 |
| Diagnosed with stroke | 2.260 | 0.000 | 1.952 | 2.616 | 2.320 | 0.000 | 2.002 | 2.690 |
| Diagnosed with diabetes | 1.316 | 0.000 | 1.149 | 1.507 | 1.330 | 0.000 | 1.160 | 1.525 |
| Diagnosed with heart disease | 1.562 | 0.000 | 1.366 | 1.787 | 1.538 | 0.000 | 1.342 | 1.762 |
| Diagnosed with asthma | 1.632 | 0.000 | 1.456 | 1.828 | 1.608 | 0.000 | 1.433 | 1.804 |
| Diagnosed with cancer | 1.449 | 0.028 | 1.042 | 2.014 | 1.476 | 0.022 | 1.057 | 2.060 |
| Diagnosed with renal failure | 1.843 | 0.000 | 1.448 | 2.346 | 1.833 | 0.000 | 1.438 | 2.338 |
| Smoking status, reference: everyday | | | | | | | | |
| Not everyday | 1.206 | 0.004 | 1.062 | 1.370 | 1.164 | 0.021 | 1.023 | 1.324 |
| Ex smoker | 1.132 | 0.049 | 1.001 | 1.281 | 1.082 | 0.215 | 0.955 | 1.226 |
| Not smoker | 0.699 | 0.000 | 0.635 | 0.770 | 0.714 | 0.000 | 0.647 | 0.786 |
| Alcohol consumption, reference: under standard | | | | | | | | |
| More than standard | 0.987 | 0.898 | 0.809 | 1.205 | 0.886 | 0.244 | 0.722 | 1.086 |

*(Continued)*

| Variables | Model 1 | | | | Model 2 | | | |
|---|---|---|---|---|---|---|---|---|
| | Odds ratio | P-Value | 95% Confidence intervals | | Odds ratio | P-Value | 95% Confidence intervals | |
| | | | Lower | Upper | | | Lower | Upper |
| No alcohol | 0.579 | 0.000 | 0.505 | 0.665 | 0.683 | 0.000 | 0.593 | 0.787 |
| Self-rated health, reference: good | | | | | | | | |
| Adequate | 2.425 | 0.000 | 2.275 | 2.584 | 2.349 | 0.000 | 2.202 | 2.506 |
| Poor | 3.375 | 0.000 | 9.504 | 11.327 | 3.361 | 0.000 | 8.557 | 10.240 |
| Household expenditure, reference: 1$^{st}$ quartile | | | | | | | | |
| 2$^{nd}$ | 0.978 | 0.579 | 0.902 | 1.059 | 0.979 | 0.611 | 0.902 | 1.063 |
| 3$^{rd}$ | 0.860 | 0.000 | 0.792 | 0.934 | 0.859 | 0.000 | 0.789 | 0.936 |
| 4$^{th}$ | 0.827 | 0.000 | 0.759 | 0.900 | 0.840 | 0.000 | 0.769 | 0.917 |
| 5$^{th}$ | 0.715 | 0.000 | 0.651 | 0.786 | 0.725 | 0.000 | 0.656 | 0.800 |
| Java | 0.611 | 0.000 | 0.573 | 0.652 | 0.713 | 0.000 | 0.603 | 0.844 |
| **District-level variables** | | | | | | | | |
| Community social capital | | | | | 0.935 | 0.050 | 0.875 | 1.000 |
| Social deprivation | | | | | 1.131 | 0.000 | 1.057 | 1.209 |
| District GDP | | | | | 0.909 | 0.002 | 0.856 | 0.966 |
| Constant | 0.039 | 0.000 | 0.032 | 0.049 | 0.065 | 0.000 | 0.037 | 0.115 |
| Variance at the individual level | | | | | 0.586 | | 0.537 | 0.640 |
| Variance at the district level | | | | | 0.095 | | 0.081 | 0.111 |
| AIC | 23,785 | | | | 23,321 | | | |
| BIC | 23,998 | | | | 23,507 | | | |

However, this relationship is unlikely to be linear; structural inequalities and uneven distribution of economic benefits can moderate these effects, even in relatively affluent districts. Future research employing longitudinal and spatial analyses could clarify causal mechanisms and identify vulnerable groups who remain at risk despite broader economic improvement.

### Individual vulnerabilities and gendered patterns of ideation

Women had significantly lower odds of reporting persistent suicidal ideation than men. This finding contrasts with the well-established gender paradox in suicide research, whereby women typically report higher levels of suicidal ideation and attempts, while men experience higher suicide mortality [30]. It also diverges from evidence from LMICs, which generally report a higher prevalence of suicidal ideation among women than men [20,31]. This discrepancy warrants cautious interpretation. One plausible explanation relates to measurement: the Riskesdas item captures recurrent or persistent suicidal or death-related thoughts, and gender differences may differ for this more severe form of ideation compared with broader measures that include occasional or fleeting thoughts. In addition, gender-differential reporting bias is likely in the Indonesian context. Strong stigma surrounding suicide, combined with gendered expectations of emotional restraint, family honour, and religious norms, may discourage women from endorsing survey items that explicitly reference suicide or wishing to be dead. As a result, the observed sex difference may reflect variation in reported suicidal ideation rather than true differences in underlying risk, underscoring the need for gender-sensitive measurement approaches in future research.

Unemployment, poverty, and divorce were also associated with higher odds of suicidal ideation, consistent with evidence that economic hardship and weakened social support increase psychological distress [5,32]. Older adults, particularly those aged 65–74 years, and individuals with chronic illnesses reported a greater likelihood of suicidal thoughts [33].

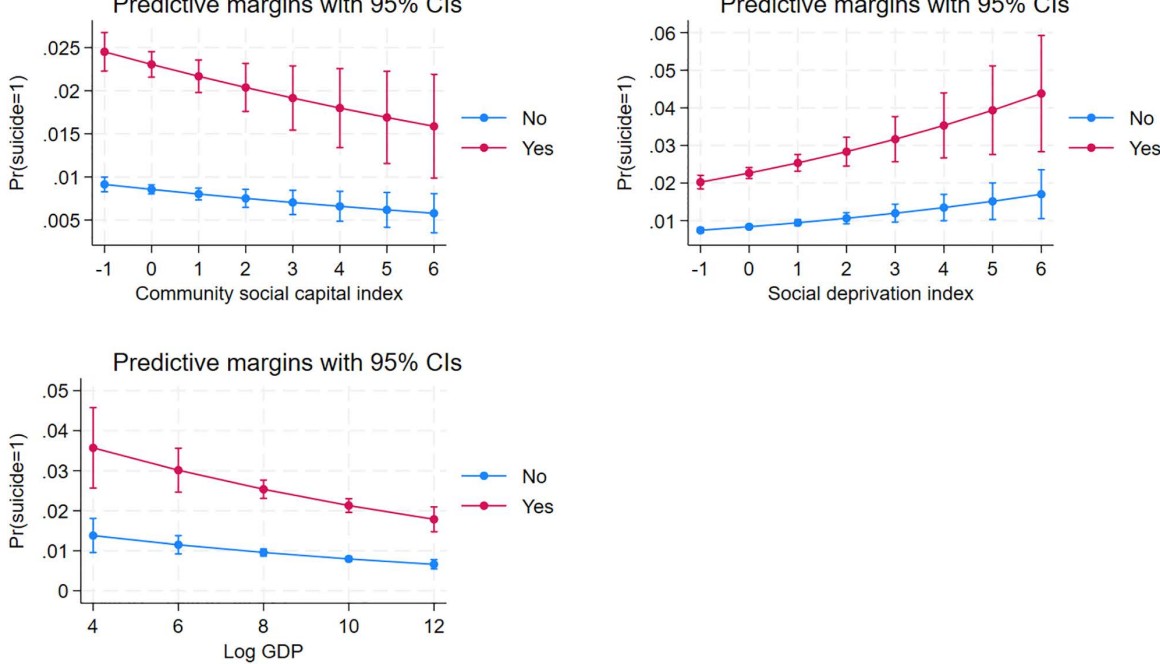

**Fig 2. Margins effect of district-level variables on suicidal ideation.**

These factors often intersect and reinforce one another: unemployment can lead to poverty and relationship breakdowns, while chronic illness may restrict work ability and increase financial dependence. The interplay of social, relational, and health-related stressors underscores the importance of integrated mental-health and social-welfare systems to mitigate cumulative risk.

### Policy pathways for suicide prevention in resource-constrained settings

This study offers several policy-relevant insights. Strengthening community social capital appears to be a promising approach to suicide prevention in resource-constrained settings. Interventions might include supporting village-level associations, faith-based or women's groups, peer-support initiatives, and volunteering programmes that build trust and participation. However, the evidence base for community-level interventions to enhance social capital and reduce suicidality remains limited, even in high-income settings, and their effectiveness in LMIC contexts is largely untested. Future research should therefore explore how such approaches can be adapted and implemented in resource-constrained and culturally diverse communities, to identify the most effective and sustainable ways to strengthen social connectedness and reduce suicide risk.

Addressing socioeconomic deprivation remains essential. Policies that expand access to education, decent employment, and social protection can reduce structural inequities that contribute to psychological distress and suicidality. Employment instability and underemployment have been consistently identified by individuals with lived experience and community stakeholders as major stressors linked to suicidal thoughts and behaviours, particularly in LMICs where social safety nets are limited [5,34]. These priorities echo global calls for multisectoral suicide prevention strategies that address social and economic determinants, as outlined in the WHO LIVE LIFE implementation guide and the Lancet–World Psychiatric Association Commission on Depression [35,36]. Strengthening job security, improving working conditions, and integrating mental health support within labour and social protection policies may therefore represent actionable and publicly supported avenues for suicide prevention.

## Methodological strengths and remaining limitations

This study provides one of the first multilevel analyses of suicidal ideation using nationally representative data in Indonesia, integrating individual and district-level information. Its large sample and robust analytical framework enhance the reliability and policy relevance of the findings.

However, several limitations must be acknowledged. The cross-sectional design precludes causal inference; associations identified here should be interpreted as correlational. Suicidal ideation was assessed using a single survey item, which may not capture the full complexity of suicidal thoughts, such as frequency, intensity, or intent [37,38]. Cultural differences in expressing or concealing suicidality may also affect the accuracy of assessment. Future research should use longitudinal or mixed-method designs with repeated measures of ideation and incorporate contextual qualitative data to better understand the social dynamics underlying suicidal behaviour.

Another limitation concerns the measurement of suicidal ideation. The study relied on a single self-reported question to assess suicidal thoughts. The study relied on a single self-reported question to assess suicidal thoughts, referring specifically to recurrent suicidal or death-related thoughts over a two-week period. While this approach is suitable for large population-based surveys, it may not fully capture the multidimensional nature of suicidal ideation, including variation in intensity, persistence, or intent. The use of a single-item measure may therefore underestimate the overall prevalence of suicidal ideation or obscure differences across subgroups, particularly among individuals experiencing occasional or emerging suicidal thoughts. Future research should incorporate validated multi-item scales or mixed-method designs to provide a more comprehensive assessment of suicidality.

## Conclusion

This study provides novel evidence on how community social capital, social deprivation, and economic context are associated with suicidal ideation in Indonesia, an LMIC where suicide remains highly stigmatised and underreported. While grounded in Indonesia's socio-cultural and economic context, these findings hold broader relevance for other LMICs facing similar structural challenges. Strengthening community social capital and addressing social deprivation are associated with lower odds of suicidal ideation. Embedding such strategies within national mental-health policies, alongside efforts to reduce socioeconomic inequality, could help mitigate distress and promote mental well-being.

Although causal relationships cannot be inferred, the mechanisms identified, such as the buffering role of social capital and the exacerbating effects of deprivation, mirror patterns observed globally. The Indonesian experience thus contributes valuable evidence for developing integrated, community-based suicide-prevention strategies adaptable to other resource-limited settings.

## Supporting information

**S1 Table. Characteristics of participants.**
(DOCX)

## Author contributions

**Conceptualization:** Sujarwoto Sujarwoto, Asri Maharani.

**Data curation:** Sujarwoto Sujarwoto, Asri Maharani.

**Formal analysis:** Sujarwoto Sujarwoto, Penny Bee, Helen Brooks, Asri Maharani.

**Investigation:** Sujarwoto Sujarwoto, Asri Maharani.

**Methodology:** Sujarwoto Sujarwoto, Asri Maharani.

**Project administration:** Sujarwoto Sujarwoto, Asri Maharani.

**Supervision:** Sujarwoto Sujarwoto, Asri Maharani.

**Validation:** Penny Bee, Helen Brooks.

**Writing – original draft:** Sujarwoto Sujarwoto, Asri Maharani.

**Writing – review & editing:** Sujarwoto Sujarwoto, Penny Bee, Helen Brooks, Asri Maharani.

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
