## [Decision Letter · Decision Letter 0]

31 Dec 2025

Dear Dr. Maharani,

Thank you for submitting your manuscript to PLOS ONE. After careful consideration, we feel that it has merit but does not fully meet PLOS ONE’s publication criteria as it currently stands. Therefore, we invite you to submit a revised version of the manuscript that addresses the points raised during the review process.

We look forward to receiving your revised manuscript.

Kind regards,

Shivanand Kattimani

Academic Editor

PLOS One

Journal Requirements:

4. We note that Figure 1 in your submission contain map images which may be copyrighted. All PLOS content is published under the Creative Commons Attribution License (CC BY 4.0), which means that the manuscript, images, and Supporting Information files will be freely available online, and any third party is permitted to access, download, copy, distribute, and use these materials in any way, even commercially, with proper attribution. For these reasons, we cannot publish previously copyrighted maps or satellite images created using proprietary data, such as Google software (Google Maps, Street View, and Earth). For more information, see our copyright guidelines: http://journals.plos.org/plosone/s/licenses-and-copyright.

**Reviewers' comments:**

Reviewer's Responses to Questions

**Comments to the Author**

1. Is the manuscript technically sound, and do the data support the conclusions?

Reviewer #1: Partly

Reviewer #2: Yes

2. Has the statistical analysis been performed appropriately and rigorously?

Reviewer #1: Yes

Reviewer #2: Yes

3. Have the authors made all data underlying the findings in their manuscript fully available?

Reviewer #1: Yes

Reviewer #2: Yes

4. Is the manuscript presented in an intelligible fashion and written in standard English?

Reviewer #1: Yes

Reviewer #2: Yes

Reviewer #1: This is a strong piece of work. The dataset is impressive. The multilevel analysis is the correct choice for this data. It handles the district-level nesting well.

However, the results seem too "clean." They contradict global trends in ways you do not fully explain. I have four major challenges for you to consider. These points test your logic and assumptions. Addressing them will make the paper much stronger.

Major Issues

1. The "Frequency" Trap

You report a suicidal ideation rate of 0.87%. This is incredibly low. The global average is often much higher. Look at your survey question again. You asked: "Have you frequently thought that you’d be better off dead?"

The word "frequently" is a major filter. It excludes everyone who has had occasional or fleeting thoughts. You are not measuring general "suicidal ideation." You are measuring "persistent death wishes." This distinction matters. It explains why your prevalence is so low. You must be honest about this limitation. You are likely missing the vast majority of at-risk people. You should reframe your results to reflect this specific definition.

2. The Gender Paradox

You found that being female reduces the risk of ideation by nearly half (OR 0.56). This contradicts the well-known "gender paradox" in suicide. Globally, women usually report more ideation and attempts than men. Men usually have higher mortality. Your data says Indonesian women think about suicide less than men.

This is highly unusual. Is it a true finding? or is it a reporting bias? Women might be less likely to admit to "frequent" death wishes due to social pressure. You accept this finding too easily. You should challenge it more aggressively in your discussion.

3. The "Gotong Royong" Assumption

You use communal labor (gotong royong) as a proxy for social capital. You assume this is always a positive thing. But in many Indonesian villages, this is a mandatory obligation. It can be a burden. High participation might mean "strict village rules" rather than "supportive neighbors."

Does forced cooperation really protect mental health? Or does it add social pressure? You treat it as a pure positive. You should consider whether it also represents social control. This is a nuance that is missing from your interpretation.

4. Causality vs. Association

This is a cross-sectional study. You cannot prove these factors cause suicidal ideation. Yet you use strong language like "determinants," "impact," and "prevent." Be careful. Poverty might cause suicidal thoughts. But suicidal thoughts might also lead to job loss and poverty. Your data cannot tell the difference. Soften your claims. Use words like "associated with" or "linked to" more consistently.

Minor Issues and Language

The writing is generally clear. However, there are awkward phrases and inconsistencies.

"Subpar alcohol consumers" (Line 263): This is incorrect. "Subpar" means "low quality." It sounds like they are bad at drinking. You likely mean "light drinkers" or "those drinking below the standard limit."

"Policeman" (Table 1): This term is dated. Use the gender-neutral "Police officer."

"Escalated" (Line 255): This is too dramatic for statistics. Use "increased."

Reviewer #2: The article provides well researched actionable evidence associating social determinants of health in Indonesia with risk of suicide, which carries high value for institutional public health management in the country.

.

Reviewer #1: **Yes:** Dr Nirmalya MukherjeeDr Nirmalya MukherjeeDr Nirmalya MukherjeeDr Nirmalya Mukherjee

Reviewer #2: **Yes:** Anoopinder Singh, MBBS, MD, FAPAAnoopinder Singh, MBBS, MD, FAPAAnoopinder Singh, MBBS, MD, FAPAAnoopinder Singh, MBBS, MD, FAPA

---

## [Author Response · Author response to Decision Letter 1]

27 Jan 2026

Comments from Academic Editors #1

Journal Requirements:

1.Please ensure that your manuscript meets PLOS ONE's style requirements, including those for file naming. The PLOS ONE style templates can be found at https://journals.plos.org/plosone/s/file?id=wjVg/PLOSOne_formatting_sample_main_body.pdf [journals.plos.org] and https://journals.plos.org/plosone/s/file?id=ba62/PLOSOne_formatting_sample_title_authors_affiliations.pdf [track.editorialmanager.com]

Authors’ response

We have checked and named the files based on the PLOS style template.

Comments from Academic Editors #2

2. We note that you have indicated that there are restrictions to data sharing for this study. PLOS only allows data to be available upon request if there are legal or ethical restrictions on sharing data publicly. For more information on unacceptable data access restrictions, please see http://journals.plos.org/plosone/s/data-availability#loc-unacceptable-data-access-restrictions [journals.plos.org].

b) If there are no restrictions, please upload the minimal anonymized data set necessary to replicate your study findings to a stable, public repository and provide us with the relevant URLs, DOIs, or accession numbers. For a list of recommended repositories, please see https://journals.plos.org/plosone/s/recommended-repositories [journals.plos.org]. You also have the option of uploading the data as Supporting Information files, but we would recommend depositing data directly to a data repository if possible.

Authors’ response

The data used in this study are not owned by the authors and cannot be shared publicly due to legal and ethical restrictions imposed by the data custodians. Individual-level data from the 2018 Indonesia National Health Survey (Riskesdas) are owned by the Indonesian Ministry of Health. Although the data are de-identified, public sharing is restricted under Indonesian data governance regulations and the terms of access specified by the Ministry of Health at the time of data approval. These restrictions are imposed by the National Institute of Health Research and Development (NIHRD), Ministry of Health, Indonesia, which serves as the data custodian.

In accordance with PLOS policy, we confirm that these data are available to qualified researchers upon reasonable request, subject to approval by the data custodians, and that the authors had no special access privileges. Requests for access to Riskesdas data can be submitted to the Ministry of Health via the NIHRD data access process. Community-level data from the Village Potential Statistics (PODES) and district-level economic indicators are owned by Statistics Indonesia (BPS) and are similarly available upon registration and approval by BPS.

We have revised the Data Availability statement to explicitly describe the nature of these restrictions, identify the responsible institutions, and provide clear guidance on how interested researchers may request access to the data:

“Data are available upon official request. The data set (RISKESDAS) can be accessed with approval by Director General of Health Policy Agency, Ministry of Health, Republic of Indonesia, at https://layanandata.kemkes.go.id/request. The authors had no special access privileges; others may obtain the data in the same manner. Community indicators from the 2018 Village Potential Statistics (PODES) from Statistics Indonesia (BPS) are available from Statistics Indonesia following registration (https://ppid.bps.go.id/?mfd=0000). All variables used in this study are described in the article and Supporting Information.”

Comments from Academic Editors #3

Authors’ response

We have revised the manuscript to ensure that the ethics statement appears only within the Methods section, in accordance with PLOS requirements. The ethics statement has been removed from any other sections of the manuscript, and the Methods section now contains the complete and final ethics information.

Comments from Academic Editors #4

4. We note that Figure 1 in your submission contain map images which may be copyrighted. All PLOS content is published under the Creative Commons Attribution License (CC BY 4.0), which means that the manuscript, images, and Supporting Information files will be freely available online, and any third party is permitted to access, download, copy, distribute, and use these materials in any way, even commercially, with proper attribution. For these reasons, we cannot publish previously copyrighted maps or satellite images created using proprietary data, such as Google software (Google Maps, Street View, and Earth). For more information, see our copyright guidelines: http://journals.plos.org/plosone/s/licenses-and-copyright [journals.plos.org].

We recommend that you contact the original copyright holder with the Content Permission Form (http://journals.plos.org/plosone/s/file?id=7c09/content-permission-form.pdf [journals.plos.org]) and the following text:

“I request permission for the open-access journal PLOS ONE to publish XXX under the Creative Commons Attribution License (CCAL) CC BY 4.0 (http://creativecommons.org/licenses/by/4.0/ [creativecommons.org]). Please be aware that this license allows unrestricted use and distribution, even commercially, by third parties. Please reply and provide explicit written permission to publish XXX under a CC BY license and complete the attached form.”

Authors’ response

Thank you for raising this issue. Figure 1 is a newly redrawn map created by the authors using openly licensed geographic boundary data obtained from openfreemap.org. The revised figure does not contain any proprietary basemaps or satellite imagery (e.g. Google Maps or similar sources).

The data used to generate the map are available under an open licence compatible with the Creative Commons Attribution (CC BY 4.0) licence, and the figure may be freely reused, adapted, and distributed under CC BY 4.0.

The figure caption has been updated to clearly state the data source and licensing information. In addition, all figures have been removed from the main manuscript file and uploaded separately as individual image files in accordance with PLOS ONE submission guidelines.

Comments from Academic Editors #5

Authors’ response

N/A

Comments from Academic Editors #6

Authors’ response

We have reviewed the references and made sure there are no retracted articles.

Comments from 1st Reviewer #1

Reviewer #1: This is a strong piece of work. The dataset is impressive. The multilevel analysis is the correct choice for this data. It handles the district-level nesting well.

However, the results seem too "clean." They contradict global trends in ways you do not fully explain. I have four major challenges for you to consider. These points test your logic and assumptions. Addressing them will make the paper much stronger.

Major Issues

1. The "Frequency" Trap

You report a suicidal ideation rate of 0.87%. This is incredibly low. The global average is often much higher. Look at your survey question again. You asked: "Have you frequently thought that you’d be better off dead?"

The word "frequently" is a major filter. It excludes everyone who has had occasional or fleeting thoughts. You are not measuring general "suicidal ideation." You are measuring "persistent death wishes." This distinction matters. It explains why your prevalence is so low. You must be honest about this limitation. You are likely missing the vast majority of at-risk people. You should reframe your results to reflect this specific definition.

Authors’ response

We thank the reviewer for this important and constructive comment. We agree that the wording and time frame of the Riskesdas item impose a substantive constraint on the definition of suicidal ideation captured in this study. The item refers to recurrent suicidal or death-related thoughts within a two-week period and therefore reflects persistent suicidal ideation, rather than occasional or transient suicidal thoughts.

In response, we have revised the manuscript to (i) correct and clarify the English translation of the survey item, (ii) explicitly define the outcome as persistent suicidal ideation in the Methods and Results sections, and (iii) expand the Discussion to acknowledge that this measurement approach likely underestimates the overall population prevalence of suicidal ideation and limits comparability with studies using broader or multi-item measures. We have also revised the interpretation of our findings, accordingly, emphasising that the identified individual- and community-level associations pertain to sustained suicidal distress, which remains a critical target for public health intervention.

Comments from 1st Reviewer #2

2. The Gender Paradox

You found that being female reduces the risk of ideation by nearly half (OR 0.56). This contradicts the well-known "gender paradox" in suicide. Globally, women usually report more ideation and attempts than men. Men usually have higher mortality. Your data says Indonesian women think about suicide less than men.

This is highly unusual. Is it a true finding? or is it a reporting bias? Women might be less likely to admit to "frequent" death wishes due to social pressure. You accept this finding too easily. You should challenge it more aggressively in your discussion.

Authors’ response

Thank you for the comment. We acknowledge that our finding, lower reported suicidal ideation among women, contrasts with the classic gender paradox in suicide research, as well as with much of the literature in other LMICs, which generally reports higher prevalence of suicidal ideation among women. In response, we have substantially revised the Discussion to explicitly recognise this divergence and to avoid interpreting the result as a straightforward protective effect:

“Women had significantly lower odds of reporting persistent suicidal ideation than men. This finding contrasts with the well-established gender paradox in suicide research, whereby women typically report higher levels of suicidal ideation and attempts, while men experience higher suicide mortality [31]. It also diverges from evidence from LMICs which generally report a higher prevalence of suicidal ideation among women than men [23, 32]. This discrepancy warrants cautious interpretation. One plausible explanation relates to measurement: the Riskesdas item captures recurrent or persistent suicidal or death-related thoughts, and gender differences may differ for this more severe form of ideation compared with broader measures that include occasional or fleeting thoughts. In addition, gender-differential reporting bias is likely in the Indonesian context. Strong stigma surrounding suicide, combined with gendered expectations of emotional restraint, family honour, and religious norms, may discourage women from endorsing survey items that explicitly reference suicide or wishing to be dead. As a result, the observed sex difference may reflect variation in reported suicidal ideation rather than true differences in underlying risk, underscoring the need for gender-sensitive measurement approaches in future research.”

Comments from 1st Reviewer #3

3. The "Gotong Royong" Assumption

You use communal labor (gotong royong) as a proxy for social capital. You assume this is always a positive thing. But in many Indonesian villages, this is a mandatory obligation. It can be a burden. High participation might mean "strict village rules" rather than "supportive neighbors."

Does forced cooperation really protect mental health? Or does it add social pressure? You treat it as a pure positive. You should consider whether it also represents social control. This is a nuance that is missing from your interpretation.

Authors’ response

Thank you for the comment. We agree that collective practices such as gotong royong should not be interpreted as uniformly beneficial and that participation may, in some contexts, reflect obligation or informal enforcement of village norms rather than voluntary mutual support. In response, we have substantially revised the Discussion to explicitly acknowledge the dual nature of community social capital in the Indonesian context:

“We found that community social capital was significantly correlated with diminished odds of suicidal ideation. Evidence from high-income and Asian contexts consistently links stronger community cohesion with reduced suicide mortality and ideation rates [6, 7, 17, 27]. In Indonesia, communities with higher participation in collective activities and stronger interpersonal trust may foster a sense of belonging and mutual support, thereby buffering the effects of stress and isolation as major risk factors for suicidality. Empirical evidence from Indonesia suggests that neighbourhood-level trust and collective engagement can be protective for mental health, including lower risks of depressive symptoms, particularly when community participation is experienced as supportive and reciprocal [28]. These findi

---

## [Decision Letter · Decision Letter 1]

19 Feb 2026

Socioeconomic and Contextual Correlates of Suicidal Ideation Among Indonesian Adults: Evidence from a Multilevel Analysis of the 2018 National Health Survey

PONE-D-25-56524R1

Dear Dr. Maharani,

We’re pleased to inform you that your manuscript has been judged scientifically suitable for publication and will be formally accepted for publication once it meets all outstanding technical requirements.

Kind regards,

Shivanand Kattimani

Academic Editor

PLOS One

Reviewers' comments:

Reviewer's Responses to Questions

**Comments to the Author**

Reviewer #1: All comments have been addressed

Reviewer #2: All comments have been addressed

2. Is the manuscript technically sound, and do the data support the conclusions?

Reviewer #1: Yes

Reviewer #2: Yes

3. Has the statistical analysis been performed appropriately and rigorously?

Reviewer #1: Yes

Reviewer #2: Yes

4. Have the authors made all data underlying the findings in their manuscript fully available?

Reviewer #1: Yes

Reviewer #2: Yes

5. Is the manuscript presented in an intelligible fashion and written in standard English?

Reviewer #1: Yes

Reviewer #2: Yes

Reviewer #1: This revision is much stronger. You have addressed the major conceptual concerns regarding the measurement of persistent suicidal ideation, the apparent contradiction with the gender paradox literature, the dual nature of gotong royong, and the need to avoid causal language in a cross-sectional design. The multilevel methods are appropriate, the sample is impressive, and the discussion is now nuanced and well grounded in the LMIC and Indonesian literature.

The remaining issues are largely editorial: (a) clean up track-change artefacts and duplicated text in the suicidal ideation measure and limitations sections, (b) correct small inconsistencies (self-rated health OR/CI values, “correlated with” vs “linked to”, “2023” vs 2018 in the Acknowledgements, residual “policemanpolice officer” strings, double full stops), and (c) perform a final language pass to harmonise UK English spelling and split a few very long sentences.

Once these minor points are resolved in a clean version, the manuscript will be ready for publication.

Reviewer #2: authors have adequately addressed comments from previous round of review, and I consider this manuscript to now be acceptable for publication

.

Reviewer #1: **Yes:** Nirmalya Mukherjee, PhDNirmalya Mukherjee, PhDNirmalya Mukherjee, PhDNirmalya Mukherjee, PhD

Reviewer #2: **Yes:** Anoopinder Singh, MD, FAPAAnoopinder Singh, MD, FAPAAnoopinder Singh, MD, FAPAAnoopinder Singh, MD, FAPA

---

## [Editor Report · Acceptance letter]

PONE-D-25-56524R1

PLOS One

Dear Dr. Maharani,

I'm pleased to inform you that your manuscript has been deemed suitable for publication in PLOS One. Congratulations! Your manuscript is now being handed over to our production team.

Kind regards,

on behalf of

Dr. Shivanand Kattimani

Academic Editor

PLOS One